# A MULTI-SCALE STRUCTURE-PRESERVING HETEROLOGOUS IMAGE TRANSFORMATION ALGORITHM BASED ON CONDITIONAL ADVERSARIAL NETWORK LEARNING

## ABSTRACT

Image transformation model learning is a basic technology for image enhancement, image super-resolution, image generation, multimodal image fusion, etc. which uses deep convolutional networks as a representation model for arbitrary functions, and uses fitting optimization with paired image training sets to solve the transformation model between images in the different sets. Affected by the complex and diverse changes of the 3D shape of the actual scene and the pixel-level optical properties of materials, the solution of the heterologous image conversion model is an ill-posed problem. In recent years, most of the proposed conditional adversarial learning methods for image transformation networks only consider the overall consistency loss constraint of the image, and the generated images often contain some pseudo-features or local structural deformations. In order to solve this problem, using the idea of multi-scale image coding and perception, this paper proposes a multi-scale structure-preserving heterologous image transformation method based on conditional adversarial network learning. First, using the idea of multi-scale coding and reconstruction, a multi-scale, step by step generator lightweight network structure is designed. Then, two image multi-scale structure loss functions are proposed, and combined with the existing overall consistency loss, a loss function for generative adversarial learning is designed. Finally, test experiments are performed on the KAIST-MPD-set1 dataset. The experimental results show that, compared with the state-of-the-art algorithms, the proposed algorithm can better suppress the local structural distortion, and has significant advantages in evaluation indicators such as RMSE, LPIPS, PSNR, and SSIM.

## 1 INTRODUCTION

Heterologous image conversion is a process of generating a modal B image of the same scene from a modal A image by constructing a pixel conversion model from modality A to modality B. Early heterogeneous image conversion requires 3D scene models and complete material information to manually design pixel conversion models. However, in practical applications, the construction of 3D models of actual scenes is difficult, the composition of materials is complex, and the acquisition of imaging characteristics of materials is difficult, which makes the solution of heterogeneous image conversion models an ill-posed problem with low efficiency and low applicability.

In recent years, deep learning has led the trend in the field of computer vision, and image translation model learning is a new solution for solving heterogeneous image translation models. Image transformation model learning is an image generation technique for image super-resolution, image enhancement, and multimodal image fusion. It uses a deep convolutional network as a representation model for an arbitrary function, and solves a transition model between images of two modalities by using a fitting optimization on a training set of paired images. The most mainstream way to learn image conversion models is Conditional Generative Adversarial Nets (CGAN) proposed by Mirza & Osindero (2014). Since CGAN can model the semantic information of images and constrain the output results, CGAN is currently used for Image transformation has become a research hotspot in the field of computer vision and the latest heterogeneous image transformation methods are based on it.

Image conversion model learning relies on a large amount of training data. This paper uses the Multispectral Pedestrian Detection (MPD) dataset published by the Korea Advanced Institute of Science and Technology (KAIST)(Hwang & Jaesik Park, 2015), referred to as the KAIST-MPD dataset, which is currently the only publicly available dataset that contains a large number of visual and infrared approximate common aperture images of multiple scenes. Although the heterogenous image pairs of this dataset have viewing angle deviation and the view field of visual image is smaller, the viewing angle deviation is small and the size of the view field remains fixed, so it is feasible to use this dataset to solve the transformation model. The research in this paper is to solve the heterogeneous image conversion model from infrared to visual based on the KASIT-MPD dataset, as shown in the figure below. The contributions of this paper are summarized as:

1.A new four-layer multi-scale encoder generator is proposed, in which the images are downsampled three times, and then they are encoded and decoded from small to large in turn. Such a network structure can make full use of the structural information in the input images;

2.Two structure-sensitive loss functions are proposed to solve the problem that L1 loss and generative adversarial loss cannot effectively constrain the structural information of a single image.

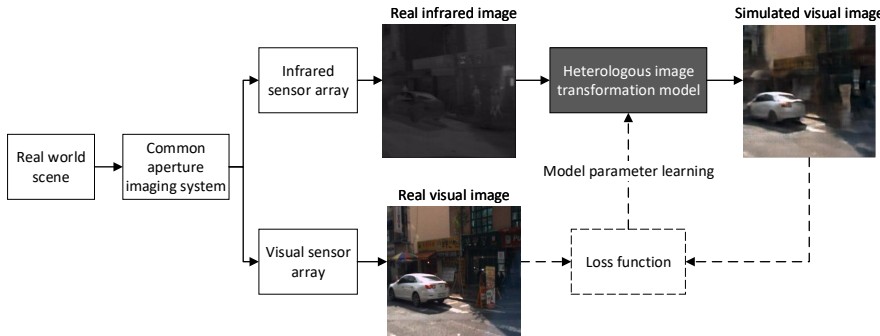

Figure 1: Heterologous image conversion from infrared to visual.

## 2 RELATED WORK

Following the development of deep learning, generative adversarial learning has also achieved rapid development. The original Generative Adversarial Nets (GAN) (Goodfellow et al., 2014) only used the feedback of the discriminator D to adjust the generator G parameters. The discriminator D replaced the method of manually designing the loss function and became a dynamic measure, but the training process of the discriminator D itself is uncontrollable and has few constraints, making it difficult to generate high-quality images. Mirza & Osindero (2014) then proposed Conditional Generative Adversarial Nets (CGAN), which sends the input images to the discriminator, and the discriminator needs to distinguish whether the two input images constitute a mapping relationship. This improvement enables the generation of the results are more purposeful.However, the details of the images generated by CGAN are still not clear enough, and there are obvious distortions in the images. In 2018, Isola et al. (2017) sorted out the idea of the image conversion model at that time, and proposed a universal pix2pix model, which uses the widely recognized ResNet (He et al., 2016) and U-net(Ronneberger et al., 2015) networks as general generators, added L1 loss, and used the PatchGAN discriminator. Different from the previous single-output discriminator, the PatchGAN discriminator will output a discrimination result for each pixel block, and finally form a discriminant surface. Experiments show that this can effectively improve the texture features of the output image, and the picture is clearer. However, pix2pix inevitably has distortion and artifacts, which cannot be effectively constrained by L1 loss and discriminative loss.The evolutionary version of pix2pix, pix2pixHD (Wang et al., 2018), is dedicated to generating larger resolution images. The model has several contributions to improve the quality of generated images: First, a new two-layer generator is built, which down-samples the input image to the lower-layer generator, and generates images from small to large; Second, the L1 loss is removed, and the Perceptual Loss (PL) (Zhang et al., 2018a) and the Feature Match (FM) loss are introduced, both of which calculate the L2 distance of

the feature block inside the discriminator. The difference is that the PL loss is used as a "third party inspection" with a discriminator pretrained on other large datasets, the mapping of the discriminator depends on the pretrained dataset, and the discriminator parameters are fixed and unchanged, while The FM loss is a "self-check", the mapping of the discriminator depends on the task training set, the parameters of the discriminator change continuously with the learning process, and the mapping also changes; Finally, the discriminator is multi-scaled, the output image is down-sampled twice, and the three images are sent to the three discriminators respectively. The experimental results show that pix2pixHD can generate clear images with resolutions above 1080P, but there are also inevitable artifacts and distortions.

At present, some domestic and foreign scholars have applied generative adversarial learning to solve heterologous image transformation models. For the application of infrared face recognition, Zhang et al. (2018b) introduced the face recognition information as part of the loss function to increase the recognizability of the generated face image. However, compared with the real visual image, the generated face image has obvious differences in facial features, which reduces the reliability of face recognition. Babu[11] added the cycle consistency constraints of infrared and visual faces on the basis of pix2pixHD, and trained both forward and reverse generators at the same time. Although the generation quality has been improved, the problems of distortion and artifacts are still unavoidable. Based on the pix2pix model, Kuang et al. (2020) introduced the PL loss and added the Total Variation (TV) loss as a regular term to make the generated image smoother, but the image distortion and artifacts were not improved.

In general, most of the existing generative adversarial methods consider texture authenticity and image consistency constraints, but ignore the loss constraints on image structure, resulting in the generated images often contain some pseudo features or local structure deformation. Therefore, how to design a loss function that pays more attention to the image structure and allow the generator to better capture the structural information of the image has become the key to overcoming the pseudo-features and local distortion of the generated image.

## 3 ALGORITHM

Since multi-scale images contain local structures at different scales, this paper proposes improvements from two aspects, generator model and loss function, to form a multi-scale structure-preserving heterogeneous image transformation scheme: First, using the idea of multi-scale encoding and decoding, a more reasonable multi-scale encoding generator with fewer parameters is proposed to fully guarantee the structural reliability of the generated image; Then, two losses sensitive to the local structure of the image are designed and introduced into the loss function to effectively suppress the deformation of the local structure of the generated image.

### 3.1 GENERATOR

The human visual system's cognition of target images is a process of multi-scale structural cognition. In the research of image super-resolution, this idea is often used to design a network of gradual synthesis from small scales to large scales to obtain high-resolution images(Guo et al., 2020). Inspired by such ideas, a four-layer generator structure is designed in this paper, as shown in Figure 2 . $Conv$ in the figure refers to the convolutional layer; The parameter $k$ refers to the width of the convolution kernel; $d$ refers to the dilation parameter, its default value is 1; $s$ refers to stride parameter, default 1; $TranConv$ refers to transposed convolution, which enlarges the length and width of the feature block to 2 times. The output feature blocks of all modules are 64 channels, and by default all convolutional layers contain instance normalization(Ulyanov et al., 2016) and activation functions. It is noted that Sitzmann et al. (2020) used the sine activation function to improve the effect of generating images. In this paper, the sine activation function with learning parameters is used as the activation function, and the formula is as follows:

$$SINE(x) = sin(\omega x + \theta) \tag{1}$$

where x is the input feature block, and angular frequency $\omega$ and phase $\theta$ are learnable parameters. Input the image into the generator, the generator firstly down-samples the image three times in a row to obtain three images of smaller size, and then starts from the image of the smallest size, the generator extracts the features and encodes them. The input of every layer of encoding network is also used to generate heterogenous images of the same scale. The generated images are up-sampled

using the nearest neighbor method and pixel-wise added to the higher-layer generated image.

The down-sampling process of the image can reduce noise and image texture, and preserve the structural information of different scales. Multi-scale coding can make full use of the structural information of each scale in the input image. A single pixel in the 1/8-size image corresponds to the 8*8 pixel area in the original image, so the bottom-up working mode of the generator is actually a synthesis process of constructing large-scale features first, and then gradually adding small-scale features. The generator proposed in this paper strengthens the correlation between multi-scale structures, and experiments in subsection4.2 will also verify the superiority of the generator.

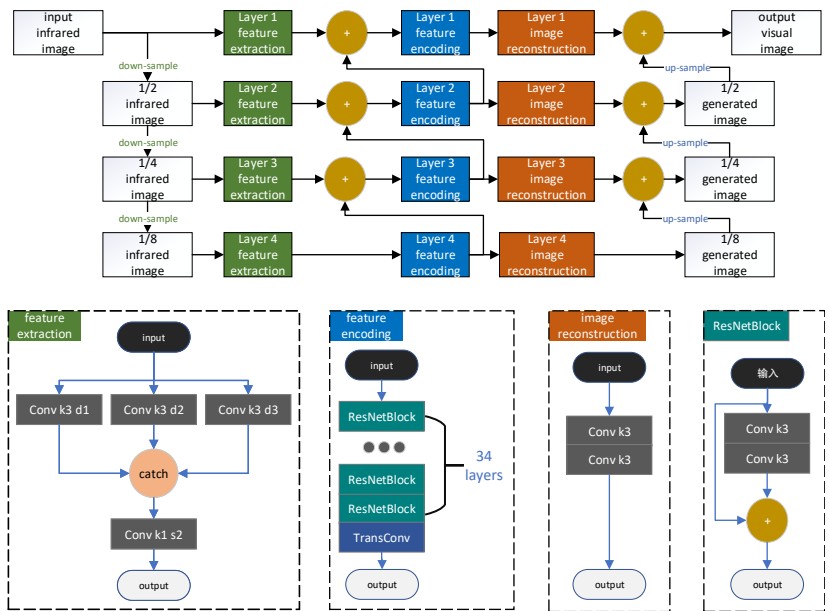

Figure 2: Model structure of generator.

## 3.2 LOSS FUNCTION

Common loss functions for image generation are used in this paper, including L1 loss, PL loss, FM loss, GAN loss and CGAN loss. L1 loss, PL loss, and FM loss are recorded as $L_{L1}$, $L_{PL}$, $L_{FM}$. Both GAN loss and CGAN loss adopt the dual form in pix2pixHD, that is, when training the generator, the generated image is True, and when training the discriminator, the real image is True, and the optimization goal is to reduce the loss at any time. The GAN loss is recorded as $L_{GAN\_G}$ when training the generator and $L_{GAN\_D}$ when training the discriminator. Similarly, the CGAN loss is recorded as $L_{CGAN\_G}$ and $L_{CGAN\_D}$ respectively. Other common loss formulas are the same as those in pix2pix and pix2pixHD, and will not be repeated in this article.

### 3.2.1 COSINE LOSS

The L1 loss calculates the pixel-level image distance, and lacks the correlation constraint between pixels; GAN and CGAN loss is based on probability to distinguish the authenticity of the image, and it is more inclined to generate high-frequency image content, resulting in severe distortion of dynamic targets. It is worth noting that when the human visual cognition system observes an image, no matter the overall color of the image is bright or dark, as long as the ratio of the values between pixels remains unchanged, people can obtain almost the same cognition, so the ratio of pixel values can represent the content information of the image data. Therefore, in this paper, each image is abstracted into a vector with the dimensions of the number of pixels × the number of channels, and then the cosine angle of the vector is used to measure the similarity between the images. The cosine

loss is defined as follows:

$$L_{COS} = -\frac{M_p(I_{vis}, G(I_{inf}))}{\|I_{vis}\| * \|I_{inf}\|} \tag{2}$$

where $I_{vis}$ represents the visual image; $I_{inf}$ represents the infrared image; $G(\cdot)$ represents the generator; $\|\cdot\|$ represents the vector modulus of image; $M_p(\cdot)$ represents multiplying the images in pixels, and then summing all pixel values of the result. When the vector angle between the generated visual image and the target visual image is 0, the cosine loss reaches the minimum value, that is, the generated image is a pixel-wise scaling of the real image. Both cosine loss and L1 loss directly calculate the image distance, but unlike the L1 loss, where each pixel is calculated independently, the cosine loss calculates the overall structure of the image, and the feedback gradient information represents the global optimization direction. It can guide the network parameter adjustment more effectively.

### 3.2.2 TOTAL VARIATION LOSS OF DIFFERENCE

Although the Total Variation (TV) loss can describe the noise level of the image and is added to the loss function as a regular term for suppressing noise, the image can only be a completely flat monochrome image when the TV loss obtains the optimal solution. Therefore, if the ratio parameter of TV loss is too large, it will inevitably lead to a lot of blurs in the generated image, and the edge information in the image is lost, which can describe the target better than color, so the introduction of TV loss is obviously flawed. In order to make the image smooth and keep the edge structure of the image better, total variation of difference (TVD) is proposed, which is defined as follows:

$$L_{TVD} = E_p(\nabla_x^2(G(I_{inf}) - I_{vis})) + E_p(\nabla_y^2(G(I_{inf}) - I_{vis})) \tag{3}$$

where $\nabla_x^2$ represents calculating the difference map of the image in the x direction, and then squaring by pixels; $\nabla_y^2$ represents calculating the difference map of the image in the y direction, and then squaring by pixels; $E_p(\cdot)$ represents calculating the average pixel value. Figure 3 shows the calculation method of the difference maps in the x and y directions in this paper.

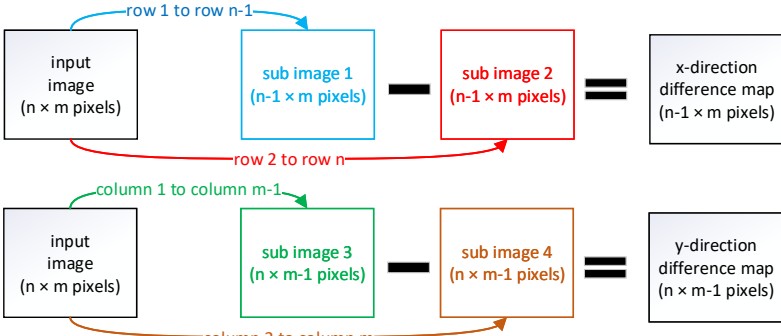

Figure 3: Get the difference map.

### 3.2.3 TOTAL LOSS FUNCTION

The total loss function of the algorithm is as follows:

$$L_G = \lambda_1(L_1 + L_{COS} + L_{TVD}) + \lambda_2(L_{GAN\_G} + L_{CGAN\_G}) + \lambda_3(L_{FM} + L_P) \tag{4}$$
$$L_D = \lambda_2(L_{GAN\_D} + L_{CGAN\_D}) \tag{5}$$

where $L_G$ represents the loss function used for training the generator; $L_D$ represents the loss function for training the discriminator; In this paper, losses are classified into three categories, and losses of the same kind use the same scaling factor: $\lambda_1$ is scaling factor of pixel class; $\lambda_2$ is the scaling factor of discriminator class; $\lambda_3$ is the scaling factor of feature distance class.

Table 1: Pseudo code of algorithm

| **Code** |
| --- |
| G = generator in this paper;
D = 5layers multi-scale PatchGAN;
C = 5layers multi-scale PatchGAN;
P = VGG19(layer 1_1 to layer 5_1);
Initialize G, D, C, P;
Import P pretraining data;
While(Do not meet training termination condition){
    Import image pairs: $I_{inf}$ and $I_{vis}$;
    Generate image $I_g = G(I_{inf})$;
    Calculate generator loss $L_G$;
    Backpropagates and updates the parameters of G;
    Generate image $I_g = G(I_{inf})$;
    Calculate discriminator loss $L_D$;
    Backpropagates and updates the parameters of D and C;
} |

### 3.3 OVERALL STRUCTURE AND PROCESS

The overall structure of the algorithm is shown in Figure 4 below, and the pseudocode of the algorithm flow is shown in Table 1. The generator G adopts the generator model in this paper. The discriminators D and C both use the multi-scale PatchGAN network, the used code of which is the same as the code in pix2pixHD, and the perception network P adopts the layer 1_1, 2_1, 3_1, 4_1 and 5_1 of pre-trained VGG19 to calculate perceived loss of output features.

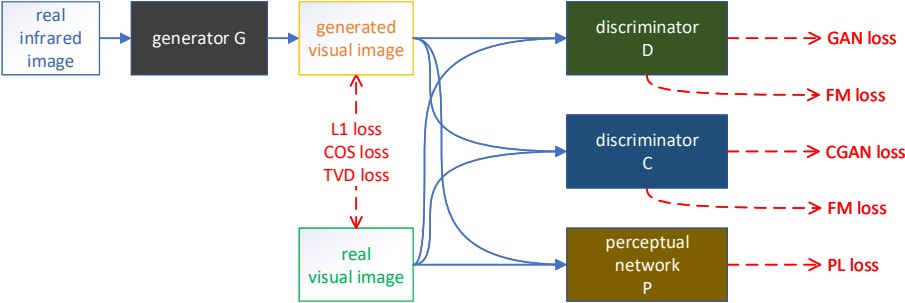

Figure 4: Overall structure of the algorithm.

## 4 EXPERIMENT

The experiments in this paper use the set01 subset of the KASIT-MPD dataset, and the 10th pair is selected as the test set for every 10 pairs of images, and the rest are used as the training set. There are 7227 pairs of infrared-visual images in the training set and 802 pairs of images in the test set. To reduce computational power consumption, all infrared and visual images were down-sampled and resized to 256×256 pixels.

In the experiment, all networks are trained for 200 epochs, using the ADAM optimizer, the parameters of which are (0.5, 0.999). The learning rate is 2e-4 initially, which in the first 100 epochs remains unchanged, but decreases linearly to 1/101 of the initial learning rate in the last 100 epochs. $\lambda_1$, $\lambda_2$, and $\lambda_3$ are respectively 100, 1, and 10. In the experiment, the first 5 layers of the VGG19 model provided by pytorch are used to calculate LPIPS, which is the same as the network structure

Table 2: Evaluation metrics for different algorithms.The bold number indicates the best result and the number with asterisk indicates the second best result.

| Net | Param(M) | The smaller the better | | The bigger the better | |
| | | RMSE | LPIPS | PSNR | SSIM |
| --- | --- | --- | --- | --- | --- |
| pix2pix | 11.37 | 10.88* | 12.96 | 17.03* | 0.62* |
| pix2pixHD | 45.59 | 13.70 | 11.76 | 16.01 | 0.60 |
| TICCGAN | 45.85 | 12.29 | 11.39* | 16.57 | 0.61 |
| PCSGAN | 11.37 | 10.93 | 12.63 | 16.98 | 0.61 |
| Ours | 10.81 | **8.61** | **10.30** | **18.11** | **0.69** |

for calculating perceptual loss, and all use the pre-training data provided by pytorch. The metrics of the generated image quality are RMSE, LPIPS, PSNR and SSIM.

## 4.1 COMPARATIVE EXPERIMENTS

The algorithm in this paper is compared with the general algorithms pix2pix(Isola et al., 2017), pix2pixHD(Wang et al., 2018) and the open source algorithms TICCGAN(Kuang et al., 2020) and PCSGAN(Babu & Dubey, 2020) for infrared to visual image conversion. The generated image comparison is shown in Figure 5, the index calculation is shown in Table 2, and the detailed comparison is shown in Figure 6. In Figure 5, the scenes in rows 2, 3, and 4 have a lot of repetition in the dataset, so all existing methods can achieve good background generation results in these scenes. But when the scene changes drastically (the scene in the first row is obtained when the car turns quickly), or when there are dynamic objects such as people and cars, the images generated by the existing algorithms suffer from a lot of distortions and false features. The reason for this image degradation is that the constraints designed in the reference algorithm do not fully utilize the pixel-related information within a single image. Algorithms that rely too heavily on the discriminator to improve image quality tend to generate more frequent information, such as static scenes, and ignore less frequent information, such as pedestrians and cars. It can be seen from the comparison that the algorithm in this paper focuses on the utilization of the information between the pixel values within a single image, better maintains the structural information, and makes the edges of moving objects clearer and more reliable. The algorithm in this paper outperforms the reference algorithm in every evaluation index, and the generator proposed in this paper has fewer parameters.

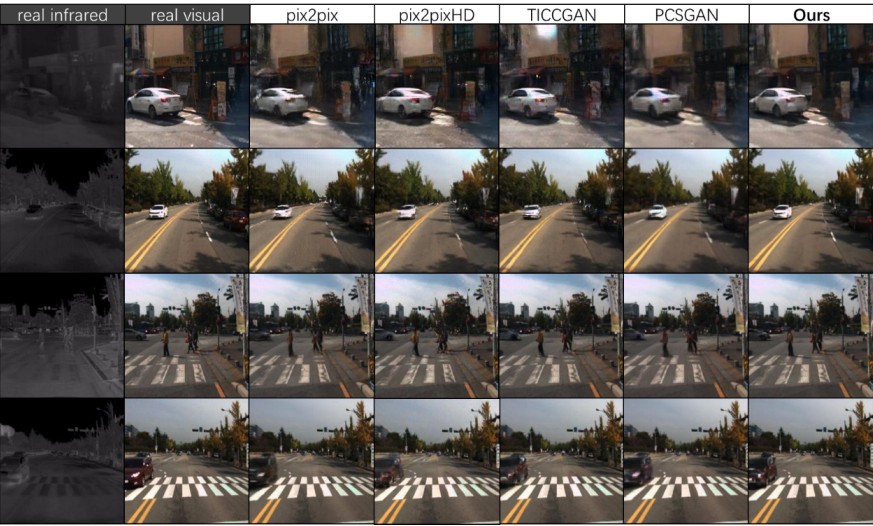

Figure 5: Comparison of multi-scene generation of different algorithms.

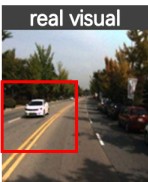
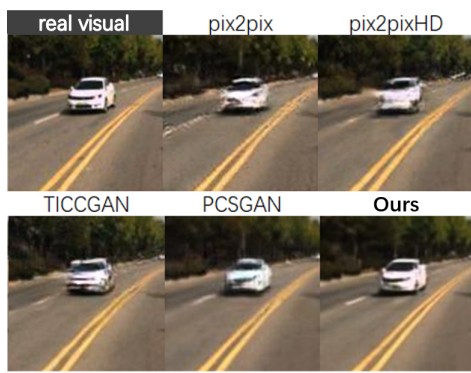

Figure 6: Comparison of vehicle generation of different algorithms.

## 4.2 ABLATION EXPERIMENT

The ablation experiments are carried out for the effectiveness of the network and loss functions proposed in this paper. The experimental results are shown in Figure 7, the evaluation indicators are shown in Table 3, and the detailed comparison is shown in Figure 8. In the ablation experiments discussing the effectiveness of the network structure, the 9-layer ResNet generator adpoted in pix2pix is used as an alternative generator G, which is similar to the network proposed in this paper in the number of parameters

According to the comparison in Figure 7 and Figure 8, the lack of any of the three improvements proposed in this paper will make the cars and people in the image more distorted, and the structural information of the image will be lost more. The evaluation indicators also prove the effectiveness of the improvements.

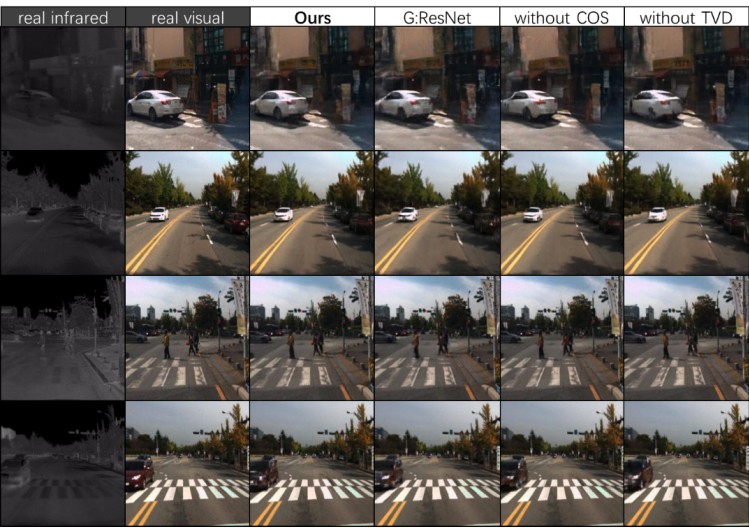

Figure 7: Comparison of Ablation Experiment.

## 5 CONCLUSION

This paper proposes a network structure for multi-scale encoding and two loss functions sensitive to image structure information. Experiments on paired infrared-visual dataset show that the algorithm proposed in this paper generates fewer image distortions and pseudo-features than the

Table 3: Evaluation indicators for ablation experiment.The bold number indicates the best result.

| Net | Param(M) | The smaller the better | | The bigger the better | |
| --- | --- | --- | --- | --- | --- |
| | | RMSE | LPIPS | PSNR | SSIM |
| **Ours** | 10.81 | **8.61** | **10.30** | **18.11** | **0.69** |
| use ResNet as G | 11.37 | 9.35 | 11.54 | 17.68 | 0.66 |
| without COS | 10.81 | 9.16 | 10.71 | 17.84 | 0.67 |
| without TVD | 10.81 | 9.01 | 10.68 | 17.9 | 0.67 |

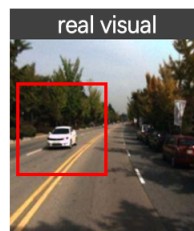
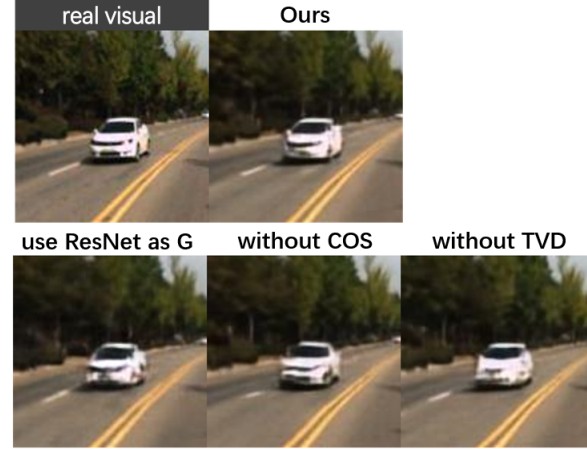

Figure 8: Comparison of vehicle generation in Ablation Experiment.

state-of-the-art algorithm, and has obvious advantages in all evaluation indicators of image quality.

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
