# OpenReview forum: "A MULTI-SCALE STRUCTURE-PRESERVING HETEROLOGOUS IMAGE TRANSFORMATION ALGORITHM BASED ON CONDITIONAL ADVERSARIAL NETWORK LEARNING"
_ICLR.cc/2023/Conference — Submitted to ICLR 2023_

### Official Review · Reviewer_o1H8 · 2022-10-24

**Confidence:** 5
**Correctness:** 2
**Technical Novelty And Significance:** 2
**Empirical Novelty And Significance:** 2
**Recommendation:** 3

**Clarity, Quality, Novelty And Reproducibility:**

The manuscript is basically clear. The technical content of the paper is correct, but some details should be added. The present paper is reproducible. The novelty of the paper is limited. The multi-scale strategy looks similar to most multi-scale feature extraction methods. The effectiveness of the proposed loss functions is not validated on other image transformation methods.

**Strength And Weaknesses:**

Strengths:
1.	The proposed loos function is capable to consider the correlation constraint between pixels.
2.	Total variation of difference (TVD) is proposed to make the image smooth and keep the edge structure of the image better.

Weaknesses:
1.	The novelty is limited. As a major contribution to the author's summary, what are the advantages of multi-scale feature extraction over existing methods? It looks similar to most multi-scale feature extraction methods.

2.	It will be better if the results of more state-of-the-art methods proposed in 2021 and 2022 are provided.

3.	More descriptions should be given in the titles of the figures and tables.

4.	The layout of the subfigures should be improved, e.g., Fig.6 and Fig.8.

5.	The results of runtime should be added in the paper.

6.	The number of references is too small. References can be further improved.

7.	The format of the tables should be revised. These tables should contain three lines.

8.	It will be better if the results of some non-reference metrics are provided.

9.	There are some grammar mistakes, e.g., “the state-of-the-art algorithm” in Conclusion should be “the state-of-the-art algorithms”.

10.	Some results of the proposed loss functions on other image transformation methods should be added to validate the effectiveness.

11.	The selection of hyperparameters used in the combination of loss functions should be analyzed.


**Summary Of The Paper:**

A multi-scale structure-preserving heterologous image transformation method based on conditional adversarial network learning is proposed in this paper. The experimental results show that the proposed algorithm can better suppress the local structural distortion and has significant advantages in evaluation indicators such as RMSE, LPIPS, PSNR, and SSIM.

**Summary Of The Review:**

See the above

---

> ### Author Response · Authors · 2022-11-08
> **Reply to Review**
>
> Firstly please let me  apologize for the distress caused by my poor writing, I'll check more closely later.
>
> Multiscale generators are indeed common, but my differences are:
> 1. I adopt down-sampling the input image multiple times, instead of down-sampling the feature block as usual, because my purpose is to allow the larger-scale information in the image to enter the network more.
> 2. I try to use 64 channels for all residual blocks, so that the parameter amount of my network is a little less than that of the classic nine-layer residual network generator, the multi-scale structure is obtained, and the dynamic objects (such as people and people) are generated with less edge distortion.It also reminded me that in the image-to-image generation task, the structure of the generator does not necessarily have to follow what it looks like in the classification task.
>
> The choice of hyperparameters is not the focus of my research, and I have limited computing resources at my disposal. Other hyperparameters refer to the public project code of other researchers.
>
> The purpose of this paper is clear, to reduce edge distortion of dynamic, low-frequency content in infred-to-visble image generation. Indeed, this discussion has limitations.
>
> Your suggestions for tables and subplots are great, thanks!
>
> Finally,thank you for your review!

---

### Official Review · Reviewer_5dFs · 2022-10-24

**Confidence:** 2
**Clarity, Quality, Novelty And Reproducibility:** Poor writing, poor quality, not much …
**Correctness:** 2
**Technical Novelty And Significance:** 2
**Empirical Novelty And Significance:** 1
**Recommendation:** 3

**Strength And Weaknesses:**

Strengths

1. Though I have seen the use of cosine loss in the literature associated with the classification or metric learning, I haven’t encountered using the same for image transformation. This can be an exciting proposal to look at.


Weaknesses

1. Poor writing standard: The paper is poorly written. Below are some examples

- Input the image into the generator, the generator firstly down-samples the image three times in a row to obtain three images of smaller size, and then starts from the image of the smallest size, the generator extracts the features and encodes them. => poorly framed sentence, written as if one is talking to another in-person

- Both GAN loss and CGAN loss adopt the dual form in pix2pixHD, that is, when training the generator, the generated image is True, and when training the discriminator, the real image is True, and the optimization goal is to reduce the loss at any time. The GAN loss is recorded as LGAN G when training the generator and LGAN D when training the discriminator
- Table 1 seems to be redundant.
- Poorly showcased results. It is extremely difficult to compare and understand the differences among the results shown in Figs. 5-7

2. Lack of novelty: Multi-scale-based network architectures are very common in image transformation and restoration. Some of these works are cited by the authors themself. Some other works, for example, includes: {Deep Laplacian Pyramid Networks for Fast and Accurate Super-Resolution CVPR 2017, Scale-recurrent network for deep image deblurring CVPR 2018, Msg-gan: Multi-scale gradients for generative adversarial networks, CVPR 2020} I do not see any novelty in terms of the design of the generator as such.

3. The paper's objective doesn’t have too much of an impact: This work aims to improve the image transformation performance specifically for the KASIT-MPD dataset. As such, the advantages of the proposed method for general image transformations are unclear.

4. Experiments can be biased: Though authors have compared with some state-of-the-art methods, there is no clear description indicating if authors have used pre-trained networks or newly trained (on the dataset under consideration) versions of these state-of-the-art methods.

Minor issue:

Section 2: Babu[11] - erroneous citation

**Summary Of The Paper:**

This work solves the problem of transforming images from infrared to real visual spectrum that too solely focuses on the KAIST-MPD dataset. To this end, the authors propose to use a multi-scale generator architecture and new loss functions such as cosine loss and total variation loss of the differences.

**Summary Of The Review:**

The work suffers from poor writing quality, lack of novelty, and too narrow scope keeping it well below the acceptance level.

---

> ### Author Response · Authors · 2022-11-08
> **Reply to Review**
>
> I'm not a native English speaker, firstly please let me  apologize for the distress caused by my poor writing, and I'll pay attention to the writing perspective later.
>
> Regarding the multi-scale generator, there are similarities with your proposed article, my understanding is:
> 1. [MSG-GAN] is to generate large images from small images to ensure the high quality of the images. The purpose of my design is to downsample the input infrared image multiple times and input it, so that the generated image can better preserve the edge consistency of images in different spectral bands.
> 2. [LapSRN] is a classic article in the field of super-resolution. The difference is that instead of just feeding in images of a certain scale, I'll feed images at every scale.
> 3. [Scale-recurrent Network for Deep Image Deblurring] uses a smaller-scale generated image as input, and I just import the pre-reconstruction features into a larger-scale generation network. For smaller-sized generated images, I just simple up-sample it and then pixel-wise add it to the larger scale generated image. In order to make the network of different scales able to have a shorter distance from the output, I used such a synthesis method.
> 4. My network does not adopt the traditional design in which people halves the length and width of feature blocks and double the number of feature channels, and all the residual blocks in my network are 64 channels, which makes my network have less parameters than the normal 9-layer resnet , but achieves multi-scale structures. The classification task makes deep learning attract the attention of researchers, and more features can construct more complex classification surfaces. But is this really necessary in image generation tasks? That's one of my motivations for trying this.
>
> My original intention was to better preserve the edge consistency of images of different spectral segments of the same scene, so I only used the KASIT dataset, not ran it on a general generation task. Admittedly, my consideration is really not enough, and I will run it on more datasets later.
>
> Thank you for your review!

---

### Official Review · Reviewer_KAPg · 2022-10-26

**Confidence:** 3
**Correctness:** 3
**Technical Novelty And Significance:** 2
**Empirical Novelty And Significance:** 2
**Recommendation:** 3

**Clarity, Quality, Novelty And Reproducibility:**

Clarity: Good to follow and easy to understand
Quality: Good. Can be accepted if more experiments are evaluated.
Novelty: Seems ad-hoc combination of existing GAN networks and losses.
Reproducibility: Seems can be reproduced.

**Strength And Weaknesses:**

Strength:
1. SOTAs on the KASIT-MPD dataset

Weaknesses:
1. Only one dataset and infrared-rgb task is evaluated.

**Summary Of The Paper:**

This paper is interesting. It summarizes varios image processing tasks as image transformation trained on adversersial networks.

**Summary Of The Review:**

This paper can be accepted if more experiments are evaluated.

---

> ### Author Response · Authors · 2022-11-08
> **Reply to Review**
>
> Thanks for the suggestion, I will try experiments on other datasets. My original intention was to build an image-to-image mapping between different spectral bands of the same scene, so only KAIST dataset was used.
>
> I am not directly piecing together the existing GAN results. To reduce possible misunderstandings,the structural information of the image I am talking about refers to the consistency of object edges in different spectral images of the same scene.My thoughts are mainly the following two:
>
> 1. How to input more structural information of a single image? The structure of the image is larger-scale feature. Each downsampling operation on the input image can retain larger-scale information, and the small-scale information is smoothed out.So I used downsampling before encoding the image instead of downsampling after encoding.The advantage of this is that features of larger size are input more times.
>
> 2. How to constrain the structural information of the image. My understanding is that in a computer, an image is just a data matrix, and the numerical size of a pixel is not the most critical information.The ratio of values between pixels and the changes between pixels are the factors that make people perceive an object, so I propose both losses.
>
> Thank you for your review of this article!

---

### Decision · Program_Chairs · 2023-01-20

**Decision:**

Reject

**Justification For Why Not Higher Score:**

The reviewers reached the consensus of reject. The paper is far from ICLR standard.

**Justification For Why Not Lower Score:**

N/A

**Metareview: Summary, Strengths And Weaknesses:**

The paper proposed a multi-scale structure-preserving heterologous image transformation based on conditional adversarial network learning. The experimental results show that the proposed algorithm can better suppress local structural distortion on standard metrics.

All the reviewers were negative of the paper: limited novelty, limited references, poor writing and very narrow scope. No additional results were provided in the rebuttal. The AC agrees with the reviewers that the paper is not ready for publication.